# Transient Complexity of *E. coli* Lipidome Is Explained by Fatty Acyl Synthesis and Cyclopropanation

**DOI:** 10.3390/metabo12090784

**Published:** 2022-08-24

**Authors:** Nikolay V. Berezhnoy, Amaury Cazenave-Gassiot, Liang Gao, Juat Chin Foo, Shanshan Ji, Viduthalai Rasheedkhan Regina, Pui Khee Peggy Yap, Markus R. Wenk, Staffan Kjelleberg, Thomas William Seviour, Jamie Hinks

**Affiliations:** 1Singapore Centre for Environmental Life Sciences Engineering, Nanyang Technological University, Singapore 637552, Singapore; 2Singapore-HUJ Alliance for Research and Enterprise, Molecular Mechanisms of Inflammatory Diseases Program, National University of Singapore (NUS), Singapore 138602, Singapore; 3Department of Microbiology and Immunology, Yong Loo Lin School of Medicine, National University of Singapore (NUS), Singapore 117456, Singapore; 4Precision Medicine Translational Research Program, Department of Biochemistry, Yong Loo Lin School of Medicine, National University of Singapore (NUS), Singapore 117456, Singapore; 5Singapore Lipidomics Incubator, Life Sciences Institute, National University of Singapore (NUS), Singapore 117456, Singapore; 6Aarhus University Centre for Water Technology, Department of Biological and Chemical Engineering, Universitetsbyen 36, 8000 Aarhus, Denmark

**Keywords:** lipidomics, cyclopropanation, *E. coli*, liquid chromatography-mass spectrometry, hydrophilic interaction liquid chromatography (HILIC)

## Abstract

In the case of many bacteria, such as *Escherichia coli*, the composition of lipid molecules, termed the lipidome, temporally adapts to different environmental conditions and thus modifies membrane properties to permit growth and survival. Details of the relationship between the environment and lipidome composition are lacking, particularly for growing cultures under either favourable or under stress conditions. Here, we highlight compositional lipidome changes by describing the dynamics of molecular species throughout culture-growth phases. We show a steady cyclopropanation of fatty acyl chains, which acts as a driver for lipid diversity. There is a bias for the cyclopropanation of shorter fatty acyl chains (FA 16:1) over longer ones (FA 18:1), which likely reflects a thermodynamic phenomenon. Additionally, we observe a nearly two-fold increase in saturated fatty acyl chains in response to the presence of ampicillin and chloramphenicol, with consequences for membrane fluidity and elasticity, and ultimately bacterial stress tolerance. Our study provides the detailed quantitative lipidome composition of three *E. coli* strains across culture-growth phases and at the level of the fatty acyl chains and provides a general reference for phospholipid composition changes in response to perturbations. Thus, lipidome diversity is largely transient and the consequence of lipid synthesis and cyclopropanation.

## 1. Introduction

*Escherichia coli* is a well-studied model microorganism, pathogen, and workhorse chassis for synthetic biology. It has a well characterised genome [1,2], proteome [3], and transcriptome [4] and, more recently, its lipidome has been studied using modern mass spectrometric techniques [5,6,7,8,9]. Nonetheless, there is currently a lack of understanding of its composition at the level of molecular species, where FA chains are quantified. This currently confounds the interpretation of exposure effects in response to drugs, organic solvents, or other kinds of lipidome stresses [8,9,10,11,12,13]. The detailed quantitative lipidome dynamics during culture growth at molecular species-resolution that would cover major glycerophospholipids (GP) of the *E. coli* lipidome is the main goal of this study.

The lipidome of *E. coli* is dominated by a few major GP classes: phosphatidylethanolamines (PE) and lysophosphatidylethanolamines (LPE), phosphatidylglycerols (PG), and cardiolipins (CL) [14,15,16] (Figure 1A). These GPs form two membranes of *E. coli’s* cell envelope, along with glycolipid A of the lipopolysaccharide layer [17]. Straight-chain saturated (SFA) or monounsaturated (UFA) fatty acyl chains are synthesised with an even number of carbon atoms (C-atoms) (Figure 1B). FAs are attached to glycerol at the sn-1 and sn-2 positions on the glycerol moiety, making cytidine diphosphate diacylglycerol (CDP-DAG) a common substrate for the biosynthesis of PE and PG, which are precursors of LPE and CL, respectively (Figure 1C) [18,19,20]. Minor fractions of PG and CL molecules are synthesised using PE [21,22], whereas the majority of CL are synthesised from two PG molecules. In turn, LPE originates from PE by the removal of one FA chain [23]. This synthetic route results in similar FA compositions across lipid classes and sets the foundation of the lipidome in *E. coli* (Figure 1C).

The cyclopropanation of UFA-containing GPs also contributes to increased lipid diversity [24,25]. UFAs in all GP classes are converted to cyclopropanated FAs (CFA) within the membranes by CFA synthase [26]. Cyclopropanation increases the C-atom count by one, resulting in FAs with an odd number of C-atoms and thus increasing the number of lipid species and contributing to lipidome complexity. While CFAs are more stable towards oxidation and stiffer that UFAs, the physiological roles of CFA may not be limited to increased oxidation and acid tolerance [24] but may extend to bilayer fluidity and bilayer role as a buffer in activated methyl cycle [27,28]. UFAs and CFAs are collectively referred to as double bond equivalents (DBE) [29].

CFA synthase expression was reported to be under the control of two promoters, one with continuous expression, and another induced at the start of stationary phase [26]. In addition, a preference of CFA synthase for cyclopropanation of FA 16:1 over FA 18:1, and CFA synthase instability due to proteolysis, have been reported [24,25,30,31].

GP biosynthesis and cyclopropanation, therefore, dynamically affect the balance between all lipid classes of the *E. coli* lipidome during culture growth [2,5,6,14,15,16,25,32,33,34,35,36,37,38]. In addition to these intrinsic dynamics, environmental stresses from temperature changes, acid shock, and antibiotic exposure affect the composition of the *E. coli* lipidome [2,7,8,9,10,11,12,13,21,39,40,41,42,43]. The details of GP biosynthesis regulation, its responses to environmental conditions, and the resulting lipidome composition changes are areas of current research [2,8,9,10,11,21,43]. However, the lipidome dynamic details are missing when the lipidome compositions were reported at single time-points during either exponential or stationary growth phases [5,6,10,11,12,13,16,36,39,44,45,46]. In addition, the compositions lack information on individual FA chains [2,8,10,16] and on lipid classes [6,30]. As a result, there are no systematic baseline studies under conditions of maximum growth rate, i.e., with glucose as a carbon source and conducted at an optimal temperature.

Here, we resolve the dynamics of the synthesis and cyclopropanation of molecular species in the *E. coli* lipidome during growth. Three *E. coli* strains were used in this study: the widely used laboratory strains BW25113 and JW0797-1, and a Bw2V strain engineered from the BW25113 to produce butanol [47]. The molecular species across three GP lipid classes [29] were quantified using tandem mass spectrometry (MS^2^) with both head group- and FA chain-based multiple reaction monitoring (MRM) after hydrophilic interaction liquid chromatography (HILIC). The bulk FA composition and the presence of CFA were determined using gas chromatography-mass spectrometry (GC-MS). The identified change over the culture growth of the individual molecular species likely accounts for the biophysical properties of lipid membranes, such as fluidity and surface charge. Our newly identified baseline for lipidome composition and dynamics under favourable growth conditions was compared to a culture grown in the presence of antibiotics—conditions relevant to both wild type and engineered strains.

To the best of our knowledge, this is one of the first reports of quantitative lipidome dynamics during culture growth at a molecular species-resolution that covers the major GP classes of the *E. coli* lipidome. This level of analysis facilitates a better understanding of lipidome dynamics. Particularly, cyclopropanation was found to be a driver of the lipidome dynamic composition, whereas GP biosynthesis was observed to be closely related to the culture growth rate. We expect the results of our study to be used as a reference for those interested in the *E. coli* lipidome dynamics, which could inform the development of new biotechnologies targeting bacterial membranes, e.g., biocides or membrane stabilising agents [48,49,50,51].

## 2. Materials and Methods

### 2.1. Culture Growth and Growth Phase Identification of E. coli Strains

The *E. coli* strains BW25113 and JW0797-1 were purchased from coli genetic stock center (Yale University, New Haven, CT, USA). Bw2V strain was a generous gift of Junsong Sun (Biorefinery Laboratory, Shanghai Advanced Research Institute, Chinese Academy of Sciences, Shanghai, China). A total of 5 × 10^5^ colony-forming units from overnight cultures of single colonies of *E. coli* strains were transferred into 20 mL of Terrific Broth medium (24 g/L yeast extract (BD), 20 g/L tryptone (BD), 4 mL/L glycerol, 17 mM KH_2_PO_4_ (Merck, Kenilworth, NJ, USA), and 72 mM K_2_HPO_4_ (Merck, Kenilworth, NJ, USA)) supplemented with 0.5% glucose (Sigma-Aldrich, St. Louis, MO, USA), and 100 mg/L ampicillin and 50 mg/L chloramphenicol for Bw2V strain [47]. The growth was monitored by measuring optical density (OD) of cultures measured at λ = 600 nm. At the beginning of culture growth (T_0_), the OD was 0.04. The 20 mL cultures were grown at 37 °C and 200 rpm. Cultures were harvested at roughly 2, 3, 4, 5, 6, and 24 h of growth for the lipid analysis, at 2, 3, 4, 5, 8, and 24 h of culture growth for BW25113, and at 2, 3, 4, 5, 6, 8, and 24 h for Bw2V to assay CFA synthase. The OD readings of cultures of each strain were not collected at the same growth times for all strains.

The three strains of *E. coli* grown here, namely, BW25113, JW0797-1, and Bw2V, were observed to reach different cell densities at a given time point, with the OD for strain Bw2V being substantially lower than that observed for both BW25113 and JW0797-1. For all strains we used a rich, chemically undefined culture medium and optimal temperatures; thus, BW25113 and JW0797-1 grew under favourable conditions. On the other hand, the growth of Bw2V was impaired, likely because it is an engineered strain and requires antibiotics for plasmid maintenance during growth. Hence, we can deem Bw2V to be grown under a stress condition. This observation shows that culture duration or OD is an unsuitable basis on which to make lipidomic comparisons.

### 2.2. Microscopy of E. coli Cells

Cells from cultures harvested in exponential (OD = 1.1 ± 0.02 for BW25113 and JW075-1 after 2.5 h of growth; OD = 0.73 ± 0.03 for Bw2V after 3.5 h of growth) and stationary phases (OD = 6.4 ± 0.8 for BW25113 and JW075-1; OD = 3.8 ± 0.2 for Bw2V after 22 h of growth for all strains) were stained with the membrane-specific fluorescent stain FM4-64 (T13320, Thermo Fisher Scientific Inc., Waltham, MA, USA). A stock solution of 10 µg/mL in 0.9% NaCl was added to the cells in media in equal volumes on a glass slide and mounted with a coverslip. The slides were incubated for 1 min on ice and immediately imaged using Leica TCS SP8 X (Leica Microsystems, Wetzlar, Germany) equipped with a white laser. Fluorescence (excitation: 558 nm, emission: 715–750 nm) and bright field images were taken in at least 9 different regions of each slide. ImageJ (v1.53c) (U. S. National Institutes of Health, Bethesda, MD, USA) was used to calculate the average area of the cell population in each condition, using the ‘analyse particles’ function.

### 2.3. The Total Lipid and Total Protein Extraction

Bacteria were harvested by centrifugation at 6000× *g* and washed three times using phosphate-buffered saline buffer (137 mM NaCl (Merck, Kenilworth, NJ, USA), 2.7 mM KCl (Merck, Kenilworth, NJ, USA), and 10 mM Na_2_HPO_4_ (Merck, Kenilworth, NJ, USA) and KH_2_PO_4_ (Merck, Kenilworth, NJ, USA)). A total of 20 µg of deuterated PE 16:0D31-18:1, 10 µg each of PG 16:0D31-18:1 and CL 14:0, and 5 µg of LPE 18:1(d7) were added as internal standards (IS) (Avanti polar lipids, Inc., Alabaster, AL, USA) to the lyophilized bacterial mass for quantification. The amounts of IS were identified based on the determined linear response range of the instrument using serial dilutions described below.

Samples were randomized to avoid the batch effects with the inclusion of empty tubes to serve as negative controls for the total lipid extraction. The total lipid extract was obtained by Bligh and Dyer method [52]. Namely, bacterial mass dissolved in 900 µL of chloroform and methanol (1:2, *v/v*) was incubated on a shaker for 1 h at 4 °C and 300 rpm. A total of 300 µL of chloroform and 250 µL of water were added to induce phase separation, and the organic phase was collected after centrifugation at 8000× *g* for 2 min. The remaining aqueous phase was re-extracted by addition of 500 µL chloroform. The combined organic fractions were dried using vacuum concentrator and dissolved in a chloroform-methanol (1:1, *v/v*) at 10 mg/mL. The resulting total lipid extract containing GPs was used in LC-MS^2^. Equal aliquots from all samples were mixed to make quantitative control (QC) sample. Response QC samples (RQC) were prepared by serial dilution of QC sample in chloroform-methanol (1:1, *v/v*).

For the identification of CFA using GC-MS, fatty acyl methyl esters (FAME) were prepared by derivatization of the total lipid extract. A total of 0.5 mg of dry lipid extract was dissolved in 100 µL toluene, 750 µL methanol, and 150 µL 8% HCl solution. The mixture was incubated at 100 °C for 1 h. After addition of 0.5 mL hexane and 0.5 mL water, the mixture was vortexed and centrifuged for 5 min at 6000× *g*. The organic phase containing FAMEs was collected. The aqueous phase was re-extracted with 250 µL hexane and combined organic phases were used for FA analysis.

The aqueous phase containing the total soluble protein was lyophilized and solubilized in an aqueous buffer with 2.5% SDS and 3 M urea. The mixture was incubated on a shaker overnight at 37 °C and 200 rpm to solubilize proteins. The insoluble fraction was removed by centrifugation at 20,000× *g* for 10 min and the supernatant was used for the quantification of the total protein using Proteoquant proteome quantification assay kit (Abcam, Cambridge, UK).

### 2.4. Identification and Quantification of Lipids

Lipids were separated using HILIC column (Kinetex 2.6 µm HILIC 100 Å, 150 × 2.1 mm, Phenomenex, Torrance, CA, USA) on an Agilent UHPLC 1290 liquid chromatography system (Agilent Technologies, Santa Clara, CA, USA). Mobile phases A (50% acetonitrile (LC-MS grade, Thermo Fisher Scientific Inc., Waltham, MA, USA) 50% 25 mM ammonium formate (Sigma-Aldrich, St. Louis, MO, USA) pH = 4.6) and B (95% acetonitrile and 5% 25 mM ammonium formate pH = 4.6) were mixed at the following gradient: 0–6 min, 99.9–75% B; 6–7 min, 75–10% B; 7–7.1 min, 10–99.9% B; 7.1–10.1 min, 99.9% B. The flow rate was 0.5 mL/min, and the sample injection volume was 2 µL.

First, GP species were identified using QC sample based on accurate mass to charge ratio and retention time (RT) of precursor ions using HILIC and a quadrupole time-of-flight mass spectrometer QToF 6550 (Agilent Technologies, Santa Clara, CA, USA) in both positive and negative electrospray ionization modes. Second, molecular species were identified using head group and FA chain fragment ions produced via collision induced dissociation. Third, head groups and FA chains were quantitated using MRM approach on triple quadrupole instrument Agilent 6495 (Agilent Technologies, Santa Clara, CA, USA). The MRM transitions are provided in the Appendix A.

Before the third step, RQC samples were quantified to establish a concentration range of the linear response of the instrument. In the third step, samples, QCs, RQCs, and blanks were used to ensure data collection accuracy. PE and LPE head group quantification was based on a neutral loss of 141 Da fragment in positive mode; and PG and CL head group quantification was based on the precursor of 153 Da fragment in negative mode. FA chains for all GPs were quantitated in negative ionization mode. Samples at 0.4 mg/mL were injected to quantify CL lipids and at 0.04 mg/mL to quantify PE, PG, and LPE lipids.

The FAME mixture was separated on Agilent 7890A instrument (Agilent Technologies, Santa Clara, CA, USA) using Rtx-1 column (l30 m × 0.25 mm × 0.25 µm, Restek, Bellefonte, PA, USA) with the following temperature gradient: 150 °C (4 min.); 4 °C/min to 250 °C (11 min). Carrier gas was helium, 20 cm/s. Injector temperature was 250 °C. Injection volume was 1.0 µL, with a 100:1 split. Detection was conducted using Selected Ion Monitoring (SIM) mode. A commercial mixture of bacterial acid methyl esters (Merck, Kenilworth, NJ, USA) was used for identification of the fatty acids based on their RT.

Our MRM-based quantitative approach did not include the low-abundance lipid classes (<<1% of total lipids), such as GP synthesis intermediates and acyl- and lyso- variants [2,14,16,35,53]. However, despite their low abundance, it does not affect the bulk bilayer properties; these molecules contributed to the results obtained using GC-MS.

### 2.5. HILIC-MS^2^ Data Processing

The raw data were processed using Agilent MassHunter Quantitative Analysis software for QQQ version B.08 (Agilent Technologies, Santa Clara, CA, USA). The areas under curve (AUC) were manually inspected and corrected according to the RT, which was around the RT of elution of IS for PE, PG, and LPE lipid classes. Sub-minute RT variation was inversely related to the MW of the precursor ions for PE and PG and was directly related to the MW for CL. MRM transitions satisfying the established criteria were used for quantification, with the coefficient of variation of QCs < 25%, signal QC/blank > 10, and Pearson R2 in RQC > 0.8. Isotopic correction of signals based on the precursor ions was performed using LICAR [54]. The sample AUC was normalized by AUC of internal standard and by protein content. Sample head group AUCs were normalized by the AUC of IS head group to quantify lipid species, and AUC of chains was normalized by a sum of IS chains and combined to quantify molecular species. Heat maps were plotted using OriginPro 2016 graphing program. MRM transitions, AUC values, and lipid concentrations are found in Excel files of Appendix A.

### 2.6. GC-MS Data Processing

The raw data were processed using Agilent MassHunter Workstation Software Quantitative Analysis Version B.08 for GCMS (Agilent Technologies, Santa Clara, CA, USA). Signal peaks were identified using ions *m/z* = 74 collected in the SIM mode and RT. For the assessment of the data quality, the same criteria as for the HILIC-MS^2^ were used. Sample peak areas were corrected for drift based on quality control samples using locally estimated scatterplot smoothing [55]. Dilutions of the commercial standard were used to obtain a calibration curve using linear fits to obtain the corrected responses of the individual FAMEs in samples. GC-MS values were presented as relative values, and therefore were not normalized to the total protein. AUCs are found in an Excel file in the Appendix A.

### 2.7. CFA Synthase Western Blot Assay

Primary *E. coli* CFA synthase-specific polyclonal antibodies were custom ordered from PolyExpress (Genscript Biotech Corp, Piscataway, NJ, USA). *E. coli* CFA synthase amino acid sequence (UniProt accession P0A9H7-1) was used for the antigen synthesis and rabbit immunization. The antibodies were purified using antigen-specific affinity. Primary Glyceraldehyde-3-phosphate dehydrogenase (GAPDH) specific monoclonal antibodies (MA5-15738, Thermo Fisher Scientific Inc., Waltham, MA, USA) and secondary anti-rabbit (#7074) and anti-mouse (#7076) antibodies from Cell Signaling Technology (Danvers, MA, USA) were used.

Harvested bacterial cell pellets were dissolved in buffer containing 50 mM HEPES (Thermo Fisher Scientific Inc., Waltham, MA, USA), 10% glycerol, 100 mM NaCl, protease inhibitor cocktail (P8465, Sigma-Aldrich, St. Louis, MO, USA), and lysozyme (L3790, Sigma-Aldrich, St. Louis, MO, USA), sonicated on ice for an hour. A supernatant was mixed with a NuPAGE LDS Sample Buffer (4X) (Thermo Fisher Scientific Inc., Waltham, MA, USA) containing β-mercaptoethanol and boiled for 10 min. Protein quantification assay (Macherey-Nagel, Duren, Germany) was used to estimate the total protein concentration.

After SDS-PAGE electrophoresis in 10% Bis-Tris 1.5-mm minigel, proteins were transferred to a PVDF membrane using iBlot transfer stack and iBlot dry blotting system (Thermo Fisher Scientific Inc., Waltham, MA, US). The membrane was blocked in PBS buffer containing 5% BSA and 0.1% Tween-20 overnight, followed by incubation with primary and secondary antibodies dissolved in the blocking buffer. After each antibody incubation, the membrane was washed five times in PBS buffer containing 0.1% Tween-20. SuperSignal West Femto Maximum Sensitivity Substrate (Thermo Fisher Scientific Inc., Waltham, MA, US) was used to develop membrane that was imaged using ImageQuant (Cytiva, Marlborough, MA, USA).

## 3. Results

### 3.1. Growth Phase as a Basis for Lipidomic Comparison between Strains

We observed that the three strains had different growth characteristics, with strain Bw2V reaching a lower cell density (OD = 3.9) compared to BW25113 and JW0797-1, which were similar to one another (OD = 5.9 and 5, respectively) (Figure 2). We were able to assign six growth phases, numbered i–vi, on the basis of the relative growth rates (Figure 2). The minimum growth rates at the beginning and end of the culture growths were assigned as lag (i) and stationary phases, respectively (Figure 2). The maximum growth rate was classified as the exponential phase (iii) and the increase and decrease in the growth rates around the exponential phase were denoted as the acceleration (ii) and deceleration phases (iv), respectively. Since the stationary phase persisted for an extended duration, it was further subdivided into early (v) and late stationary phases (vi). 

The sampling points are clearly identified in relation to the growth phase in Figure 2. The main difference between the two conditions, aside from the absolute cell density, is that a shorter exponential phase along with an extended deceleration phase was observed for the Bw2V strain grown under antibiotic stress.

### 3.2. Lipidome Dynamics Is Determined by Synthesis and Cyclopropanation

Out of all the possible molecular species in the PE and PG classes, the most abundant molecular species in the acceleration phase were dominated by combinations of FA 16:0, FA 16:1, and FA 18:1 (Figure 3A). Similar relative abundances of the major PE and PG molecular species are demonstrated on the heatmap in Figure 3A and arise because CDP-DAG is a common substrate for PE and PG synthesis.

The dynamics in the *E. coli* lipidome can be understood with reference to PE (the most abundant lipid class). The main observations with PE remain true with PG, as the FA chain compositions of both PE and PG species are similar. Additionally, these lipids are biological precursors to other major lipid classes (e.g., LPE and CL). The abundance, and chain composition of the LPE molecular species reflects the abundance of PE molecular species that were used for their biosynthesis, as exemplified by the high abundance of LPE 16:1 (Appendix A). It is very likely that the same FA distribution also occurs in the CL class, although we did not reliably quantify it. 

### 3.3. Cyclopropanation Dynamics of Lipids with Unsaturated Chains and Transient Diversity

The PE and PG species form three groups based on the number of DBEs (DBE = 0, 1, or 2; Figure 3B–D), the CL species form five groups (DBE = 0 to 4), and the LPE species form two groups (DBE = 0 or 1; Appendix A). Lipid species that are related by cyclopropanation can be identified within the DBE-containing groups. For example, the single UFA-containing molecular species PE 16:0_16:1 was converted to CFA-containing PE 16:0_17cy; hence, they are cyclopropanation-related. Their cyclopropanation-driven abundance dynamics are shown in Figure 3E, wherein the concentration of PE 16:0_16:1 gradually decreased 32-fold and PE 16:0_17cy increased seven-fold between the acceleration and the late stationary phases. Figure 3C,F show the dynamics of four molecular species related by the cyclopropanation of PE 16:1_18:1.

The UFA/CFA ratios for the representative groups of species decreased nearly 200 times during culture growth, indicating a gradual accumulation of cyclopropanated molecules (Figure 4A).

The plot of the UFA/CFA ratios of 20 groups related by cyclopropanation (accounting for 44 molecular species), as a function of their combined abundance UFA+CFA, shows that the cyclopropanation of FA 16:1 precedes that of FA 18:1 (Figure 4B). In the deceleration growth phase (iv_2_), when the CFAs are highly abundant, the cyclopropanation groups appeared separated along the UFA/CFA axis based on the UFA chain length (Figure 4B). Specifically, all FA 16:1-based groups had a UFA/CFA < 1, indicating a tendency towards cyclopropanation for FA 16:1-containing species. In comparison, all FA 18:1-based groups had a UFA/CFA > 1, with the exception of the PG 16:0_18:1-based group. We can conclude that the cyclopropanation of FA 16:1 precedes that of FA 18:1. For example, molecules containing both FA 16:1 and FA 18:1 chains at the same level of abundance (e.g., PE 18:1_16:1 and PG 18:1_16:1) displayed UFA chain length-dependent cyclopropanation, with the majority of FA 16:1 being cyclopropanated, while the majority of FA 18:1 remained unsaturated (Figure 4B). This cyclopropanation difference between the FA 16:1 and FA 18:1 chains can be observed in the growth phases of PG 16:1_18:1 (Figure 4A).

The negative slopes of the linear fits for the two groups comprising molecules with FA 16:1 and FA 18:1 indicate that the cyclopropanation is related to the abundance of molecular species during culture growth (Figure 4C). The decreasing y-intercepts indicate that the cyclopropanation of both FA 16:1 and FA 18:1 proceeds gradually throughout all the growth phases (Figure 4D). A total FA quantification using GC-MS further supports our results with respect to the gradual cyclopropanation across growth phases of FA 16:1 ahead of FA 18:1 across the whole lipidome (Appendix A).

We detected CFA synthase in the cultures from the acceleration through the late stationary growth phases (Figure 4E).

### 3.4. Dynamics of the Saturated to Unsaturated Lipid Ratio—A Surrogate for Fluidity

The optimal membrane fluidity is essential for the correct metabolic functioning of microorganisms [56]. *E. coli* regulates membrane fluidity by varying the ratio of SFA to UFA at the stage of FA biosynthesis in response to, for example, growth temperature [57,58].

An SFA/DBE ratio of around 0.5 was observed at the outset of growth for both BW25113 and JW0797-1 (Figure 5A), which gradually increased to about 0.9 as the growth rate decreased when the cultures reached the late stationary phase. The GC-MS results independently confirmed such a trend, albeit at higher SFA/DBE ratios (Figure 5A).

The SFA/DBE ratios of PE and PG lipid classes increased in a manner that reflected the dynamics revealed in the combined bulk FA chain analysis (Appendix A). However, for LPE, significantly lower SFA/DBE ratios were observed. This likely results from LPE biosynthesis, where the FA 16:0 chains of PE are removed for lipid A acylation, which results in proportionally more DBE in LPE as a class trait relative to the other GP classes [57,59].

The increase in the SFA/DBE ratio can be ascribed to the changes in the abundance of individual molecular species. Since PE is the most abundant lipid class in *E. coli* (>70% of whole lipidome), an examination of the changes in the abundance of individual PE molecular species is a good starting point to understand what controls the SFA/DBE ratio. Instead of using the individual DBE-containing PE molecular species that show a large abundance variation due to cyclopropanation, the cyclopropanation groups can represent DBEs. For example, the cyclopropanation group originating from PE 16:1_18:1 (comprising PE 16:1_18:1, PE 17cy_18:1, PE 16:1_19cy, and PE 17cy_19cy) decreased two-fold from 20% to 10% in relative abundance throughout the growth period (Appendix A). A similar decrease was observed for another cyclopropanation group containing two DBEs based on PE 16:1/16:1. Conversely, the molecular species originating from PE 16:0_18:1, and SFA-containing molecular species PE 14:0_16:0 and PE 16:0/16:0, increased in abundance. The most abundant PE cyclopropanation group, based on PE 16:0_16:1, remained at around 40% of the total PEs across all growth phases (Appendix A). Thus, the change in the SFA/DBE ratio can be attributed to particular molecular species that varied in abundance, such as PE 16:0_18:1, PE 16:1_18:1, PE 16:1/16:1, PE 14:0_16:0, and PE 16:0/16:0. In contrast, the abundance of the molecular species PE 16:0_16:1 and PE 18:1/18:1 remained constant. The PG species displayed dynamics similar to PE.

### 3.5. Membrane Surface Charge Dynamics

Next, we assessed the head group compositional changes as an indication of the average surface charge of the *E. coli* lipid membranes. The molar charge ratio of zwitterionic PE and LPE to anionic PG varied in the 2.5–3.75 range throughout growth (Figure 5B), equating to around 21–29% of anionic charges in the membrane, respectively. This agrees with earlier studies that estimated anionic head groups to constitute 15–20% of the total lipids in the K12 strain [58]. In rich growth media and at an optimal growth temperature, the net surface charge decreased (Figure 5B). *E. coli* sense [60] and tolerate changes in lipid compositions [10,16], which in turn affects the surface charge. An increase in the anionic lipid content above 30% also decreases growth rates and stress responses [34]. The modification of surface charges represents an important mechanism of cationic antimicrobial peptide resistance in a variety of microorganisms [61]—conditions not relevant to the present study. However, modifications that would involve LPS layer [62,63] or post-synthetic modifications of lipids, such as an addition of amino acids [64], were not assessed in our study. Our results enable the direct assessment of the contribution of lipidome changes to microbial surface charges.

### 3.6. E. coli Envelope Dynamics

We estimated the total lipid abundance to be around 3–4.5 millimoles of lipid per gram of total protein (mmol/g) (Figure 6A). The abundance of lipids as a proportion of the total protein decreases across growth phases until the late stationary phase where the abundance rebounds. To relate the observed changes in the total lipid abundance to changes in the bacterial envelope, we estimated the bacterial cell size using confocal microscopy. We observed a larger mean projected area of the bacterial cells in the exponential phase (3.2 µm^2^ and 2.7 µm^2^ for BW25113 and JW0797-1, respectively) relative to the cells in the late stationary phase (2.5 µm^2^ and 2.1 µm^2^ for BW25113 and JW0797-1, respectively) (Figure 6B), in agreement with earlier studies [39,65]. In addition, the signal from the lipophilic fluorescent membrane dye (FM 4–64), under the same staining conditions, was observed to be brighter for bacteria in the stationary phase in comparison to those in the exponential phase (Figure 6C).

We determined the lipid class compositions in the *E. coli* lipidome to be as follows: PE 69–77%, PG 21–29%, and LPE 1–2%. These compositions agree with earlier studies [5,7,23,36].

### 3.7. Effects of Antibiotics

The Bw2V strain differs from its parental strain BW25113 by the presence of two plasmids that contain butanol synthesis genes [47]. The Bw2V cultured in the presence of ampicillin and chloramphenicol, antibiotics needed to retain the plasmids, grew to lower OD values (Figure 2B). The presence of antibiotics in the media adversely affected Bw2V’s growth rate from the onset of culture growth (Figure 2B). A decrease in the growth rate was accompanied by an extended deceleration phase for the Bw2V strain. The maximum growth rate of Bw2V was nearly five times lower than for the strains grown in the absence of antibiotics but its cell size was comparable to those of the strains grown in the absence of antibiotics (Figure 6B). Therefore, the presence of antibiotics reduced the growth rate but did not influence the cell size dynamics, with the latter finding in agreement with earlier reports on *E. coli* growth under chloramphenicol [66].

The most striking inter-strain difference was the substantially increased SFA/DBE ratio for Bw2V. The higher SFA/DBE = 0.9 in the exponential phase displayed by Bw2V, compared to SFA/DBE = 0.6 for BW25113 and JW0797-1 (Figure 5A), can be attributed to lower levels of PE 16:1_18:1, PE 18:1/18:1, and PE 16:1/16:1, and higher levels of PE 16:0/16:0 and PE 14:0_16:0 in Bw2V compared to the BW25113 and JW0797-1 strains (Appendix A). The SFA/DBE ratio in Bw2V increased to 1.3 in the late stationary phase and followed similar dynamics as observed for the BW25113 and JW0797-1 strains (Figure 5A). This quantitative difference in the PE molecular species in Bw2V likely originates as a response to antibiotic stress from the beginning of culture growth, which involves the regulation of FA synthesis. The increased SFA/DBE due to antibiotic stress did not prevent a further increase in the SFA/DBE that is usually observed upon transition to the stationary phase. The differences in the total lipid amounts in the strain Bw2V were found to be insignificant to the other strains in phases with active culture growth, which are acceleration, exponential, and deceleration phases. Significantly lower total lipid amounts in the Bw2V strain were observed in subsequent growth phases, such as the late deceleration and the late stationary phases, in comparison with the BW25113 strain (Appendix A). The zwitterionic/anionic ratio in Bw2V was similar to the BW25113 and JW0797-1 strains (Figure 5B), which is in agreement with previous studies where antibiotics were not observed to induce modifications of the membrane surface charge [40,67].

The cyclopropanation in Bw2V proceeded similarly to the other strains, showing gradual lipidome cyclopropanation in a chain length- and concentration dependent manner (Figure 4C,D). The higher UFA/CFA intercept values for both the FA 16:1 and FA 18:1 groups in the deceleration phase for Bw2V compared with two other strains (Figure 4D) likely reflected the extended culture growth (Figure 2) that delayed the overall cyclopropanation. Nevertheless, the UFA/CFA intercept values in the FA 16:1-containing group in the late stationary phase reached values comparable to the other strains. Since the extent of cyclopropanation, expressed as UFA/CFA, depends on both lipid synthesis and cyclopropanation, we conclude that it is likely the extended culture growth of Bw2V that was accompanied by lipid synthesis that resulted in increased values of UFA/CFA intercept. Thus, the cyclopropanation was not directly affected by antibiotics because an extent of cyclopropanation comparable to other strains was achieved in the late stationary phase, when growth ceased (Figure 4D).

## 4. Discussion

Bw2V is a genetically modified strain that needs to be grown with antibiotics to maintain the plasmid [44]. To distinguish the effects of nutrient depletion and antibiotic stress on the lipidome of strains with different growth rates, we compared strains based on growth phases rather than OD or on a temporal basis, as normally used in studies with one strain and one condition [9]. Our approach allows for a comparison of different strains that grow in different conditions, such as engineered strains that produce butanol, mutant, or pathogenic strains [9,11,12,16,47].

The abundance of UFA-containing molecular species in the *E. coli* lipidome is decreased by cyclopropanation along with a concomitant increase in the CFA-containing molecular species during culture growth. We showed this transformation in the PE, PG, and LPE lipid classes that we quantified (Figure 3B,C and Appendix A), and a recent study reported cyclopropanated CLs [9].

The preference of CFA synthase for a shorter substrate, particularly for FA 16:1 over the FA 18:1 chains in our study, was reported earlier for combined FA samples [25,30,31] and our results confirm this phenomenon at the molecular-species level. The bias towards FA 16:1 cyclopropanation relative to FA 18:1 may reflect the difference in hydrophobicity related to the FA chain size [68]. Shorter hydrophobic chains present a smaller thermodynamic barrier for binding to both the bilayer and CFA synthase [26]. This small difference in binding energies may lead to faster cyclopropanation kinetics and result in exponential differences between FA 17cy and FA 19cy cyclopropanation products. The difference between the cyclopropanation of FA 16:1 and FA 18:1 has previously been attributed to the involvement of an alternative sigma factor in the stationary phase [69], reflecting a more complex mechanism.

The observed consistent and gradual increase in CFA abundance indicates constant CFA synthase activity from the acceleration to the late stationary growth phases. This is further supported by the presence of CFA synthase in all the assayed growth phases, suggesting that cyclopropanation occurs in all growth phases, and not only in the deceleration and stationary phases, as previously stated [24,26,30,32].

Viewing the lipidome in terms of cyclopropanation groups simplifies the interpretation of lipidome complexity and raises interesting questions about the function of individual lipids, particularly those in the transitory state and that are only partially cyclopropanated. Such insights into lipid dynamics could be important diagnostic starting points from which perturbation studies could be framed. Additionally, cyclopropanation is an important process in biotechnology. The advantages of CFAs such as their low melting temperature and resistance to oxidation make them superior biodiesel alternatives to SFA and UFA [70], and understanding CFA synthase activity and regulation could have applications in biotechnology.

Based on the insights gained from the quantification of molecular species, cyclopropanation dynamics during culture growth can be summarised as follows: (1) during the acceleration phase, the composition of the lipidome is dominated by lipids with UFA chains; (2) during the late stationary phase, the composition of the lipidome is dominated by lipids with CFA chains (i.e., they have undergone complete cyclopropanation); and (3) the period between exponential and late stationary phase is characterized by the greatest lipid diversity and comprises a mixture of UFA and CFA and species that are partially cyclopropanated.

Our observed SFA/DBE ratios agree with previously reported SFA/DBE ratios of around 1.2 for an *E. coli* K-12 strain grown to the late exponential phase at 37 °C without antibiotics [58]. The observed increase in the SFA/DBE ratio is indicative of a tendency towards decreased membrane fluidity and agrees with a known response to starvation that occurs in the stationary phase [71,72]. Therefore, the SFA/DBE ratio is a way to infer the dynamics of a biophysical parameter, i.e., membrane fluidity, during growth. When combined with other tools, such as the UFA/CFA ratio and growth phase analysis, this could be used to more adequately interpret microbial lipidomic datasets.

While an SFA/DBE ratio based on FA chains provides an assessment of fluidity, it does not discriminate between different pairings of FA chains. FA chains, comprising asymmetric PE 16:0_16:1 and PG 16:0_16:1, are overrepresented at the expense of PE 16:0/16:0 and PE 16:1/16:1 (Appendix A). This bias and its biological significance may be related to either membrane–associated proteins [73] or bilayer biophysical properties and warrants further studies at the level of the regulation of phospholipid biosynthesis, studies that are outside the scope of this work.

The change in the bacterial cell size is directly proportional to both the envelope and the soluble protein fraction that we employed to normalise quantitated lipids. However, the former scales with the surface, while the latter scales with the volume. As cells increase in size during growth, the surface to volume ratio decreases. Therefore, an increase in bacterial cell volume during the exponential phase alone may account for the apparent decrease in the total lipids when normalized to the total soluble protein. The difference in brightness of the lipophilic dye indicates that the lipid fraction of the bacterial envelopes is higher during the stationary phase than the exponential phase. The corollary of this, i.e., an increase in the membrane protein concentration during the exponential phase, has been reported previously [74,75,76,77]. The systematic bias introduced by this normalisation procedure suggests the need to incorporate the dynamic nature of the *E. coli* cell mass and the lipid fraction within the envelope when using absolute quantities and further strengthens the use of the ratio metric approaches we have used to rationalise lipidome dynamics.

So, while the composition of the *E. coli* lipidome appears complex because of a large number of possible molecular species over the main lipid classes due to numerous combinations of FAs during GP synthesis (Figure 1C), it is possible to arrive at a simplified understanding of lipidome complexity by first understanding the way that the cyclopropanation of unsaturated lipid species transiently influences the lipidome (Figure 4). Following this insight, SFA-containing species can easily be folded into the analysis to assess the parameters related to the FA chain composition, such as the fluidity of membranes, and the head group-related parameters, such as the membrane surface charge (Figure 5). Finally, with a firmer understanding of the lipidome, it is possible to explore its relationship to bacterial envelope size and composition (Figure 6).

Our study shows that *E. coli* lipidome is not a snapshot but is dynamic. This dynamic nature is important for comparing studies, to interpret lipid compositions, and to identify strategies for intervention and membrane engineering.

## 5. Conclusions

The dynamics of the *E. coli* lipidome under favourable conditions have not been described at the level of fatty acyl composition until recently [9]. Our quantitative study of the most abundant lipid classes in *E. coli* revealed that lipidome dynamics are related to culture growth. The growth rate affects FA biosynthesis, which in turn determines the balance of SFA to UFA, the length of FA chains, and their pairing. A decrease in the growth rate by either nutrient depletion (exemplified by the stationary phase of BW25113 and JW0797-1 strains) or by the presence of antibiotics in the medium of Bw2V strain results in an increased SFA/DBE ratio.

The observed gradual cyclopropanation of UFA chains appears to be an essential post-synthetic modification through all growth phases that is based on the abundance and length of UFA chains. The continual process of cyclopropanation causes lipidome diversity to be at its greatest during the exponential phase due to the co-existence of UFA-containing molecules and their cyclopropanated and partially cyclopropanated counterparts. Thus, much of the observed diversity in the *E. coli* lipidome is transient—essentially arising from the presence of molecules that have only partially reached their final cyclopropanated state. Most importantly, we observed that cyclopropanation, a process commonly associated with the stationary phase, occurs continuously throughout growth.

Our findings provide a baseline dataset for comparative lipidomics with respect to *E. coli* and our approach establishes a general framework for future comparative lipidomic studies that seek to assess the impacts of a given treatment in the lipidome, an approach that is likely generally applicable to bacteria. Studies on lipidome dynamics will increase the utility of microbial lipidomics in synthetic biology and biotechnology and invite a broader appreciation of the important field of microbial lipidomics.

## Figures and Tables

**Figure 1 metabolites-12-00784-f001:**
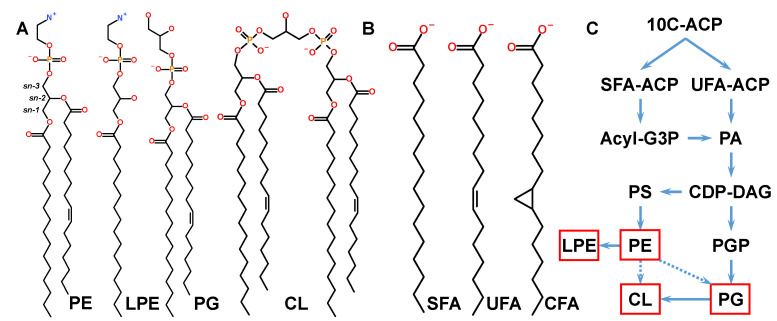
(**A**) Structures of lipid class representatives in physiological conditions with head group charges. PE 16:0_16:1 and LPE 16:0 head groups are zwitterionic. PG 16:0_16:1 and CL 16:0_16:0_16:1_16:1 headgroups are anionic, with −1*e* and −2*e* charges, respectively. Stereospecific centres are labeled “*sn*” on the glycerol residue of PE. (**B**) FA fragments with 16 C-atoms quantified in LC-MS^2^. SFA is represented by FA 16:0, UFA by FA 16:1, and CFA by FA 17cy. (**C**) Schematic biosynthesis of *E. coli* lipids. Quantified lipid classes are indicated by red frames. Minor routes of PG and CL biosynthesis from PE are indicated by dashed arrows. Abbreviations: 10C-ACP—ten carbon atom acyl-acyl carrier protein; G3P—glycerol-3-phosphate; PS—phosphatidylserine; PG—phosphatidylglycerol phosphate.

**Figure 2 metabolites-12-00784-f002:**
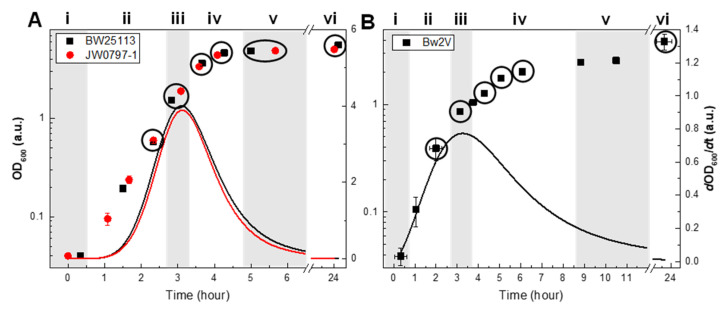
Growth rate-derived growth phases of *E. coli* strains. The optical densities in cultures of the BW25133, JW0797-1 (**A**), and Bw2V (**B**) strains are indicated by points (left ordinate scales), and the growth rates are indicated by lines (right ordinate scales). Growth phases are indicated by roman numerals and distinguished by grey backgrounds: i—lag phase; ii—acceleration (early exponential) phase; iii—exponential phase (steady state); iv—deceleration (late exponential) phase; v—early stationary phase; vi—late stationary phase. Six points indicated by circles on growth curves of each strain were used for lipid quantitation. Each point on a growth curve is an average of three biological replicates with standard deviation.

**Figure 3 metabolites-12-00784-f003:**
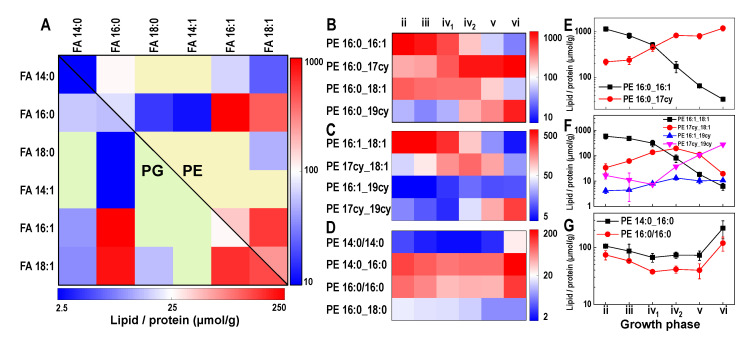
(**A**) PE and PG molecular species in the acceleration phase of the BW25113 strain are represented as combinations of two FA chains. The chain combinations that were under quantification limits are indicated by yellow and green colors for PE (top right half) and PG (bottom left half), respectively. (**B**) The abundance dynamics of the two most abundant PE molecular species with one UFA and their cyclopropanated variants. (**C**) The abundance dynamics of the most abundant molecular species with two UFAs, with two cyclopropanation intermediates and a double cyclopropanated variant. (**D**) The abundance dynamics of PE molecular species with SFA chains. (**E**–**G**) Abundance dynamics of selected PE molecular species from (**B**,**D**). All quantified molecular species are provided in Appendix A. Growth phases are indicated by roman numerals using notation introduced in Figure 2.

**Figure 4 metabolites-12-00784-f004:**
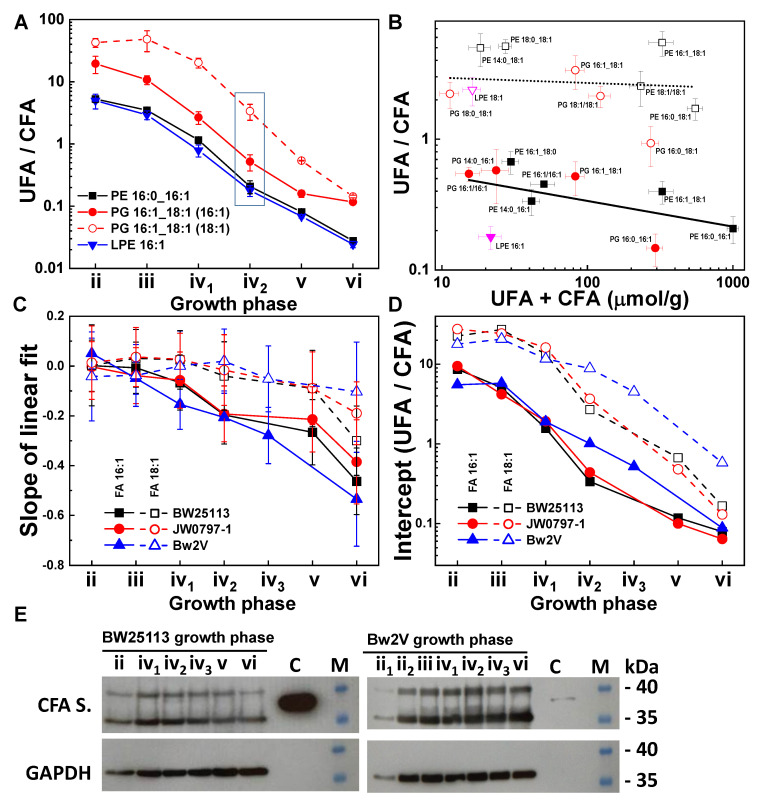
Cyclopropanation dynamics of *E. coli* lipidome. (**A**) Decreasing UFA/CFA ratios in representative cyclopropanation groups of PE, PG, and LPE lipid classes in the BW25113 strain during growth. The values in the blue square from the deceleration phase (iv_2_) were used in Figure 5B. (**B**) UFA/CFA ratios as a function of their combined abundance (UFA+CFA) from the BW25113 strain in the deceleration phase (iv_2_). The black line shows a linear fit to data points with FA 16:1 chains (solid symbols), whereas the dotted line shows a linear fit to data points with FA 18:1 chains (open symbols). (**C**) Slopes of linear fits to UFA/CFA ratios from molecular species containing FA 16:1 and FA 18:1 from growth phases. (**D**) Y-intercepts of linear fits during the culture growth were measured at 100 µmol/g of UFA+CFA. (**E**) Western blots of CFA synthase detected in growth phases of *E. coli* BW25113 and Bw2V strains. The antigen (C) was used as a positive control and GAPDH as a loading marker along with the protein molecular weight markers (M). The samples from the earliest cultures in the acceleration phases, ii of BW25113 and ii_1_ of Bw2V strains, displayed insufficient loading for 20 mL cultures at OD < 1.

**Figure 5 metabolites-12-00784-f005:**
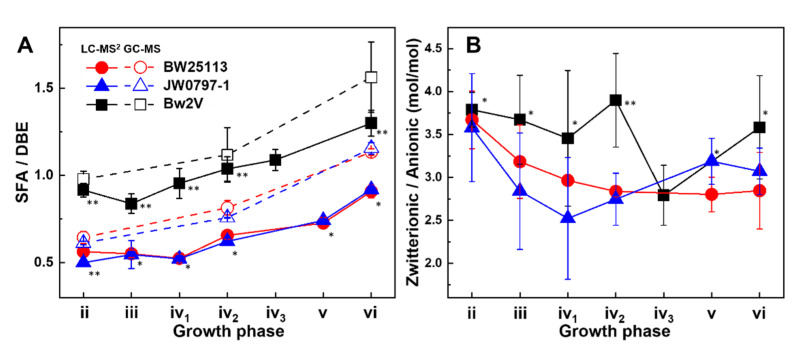
Averaged biophysical dynamics of *E. coli* lipidome. (**A**) The molar ratios of SFA to DBE chains in *E. coli* strains quantitated using LC-MS^2^ (solid symbols) and GC-MS (open symbols). * is for *p* > 0.05 and ** for *p* < 0.05 using two-tailed two-sample equal variance *t*-test between BW25113 and Bw2V or BW25113 and JW0797-1. (**B**) The molar ratio of zwitterionic to anionic head group charges ([PE+LPE]/[PG]) in *E. coli* strains. * is for *p* > 0.05 and ** for *p* < 0.05 using one-way ANOVA.

**Figure 6 metabolites-12-00784-f006:**
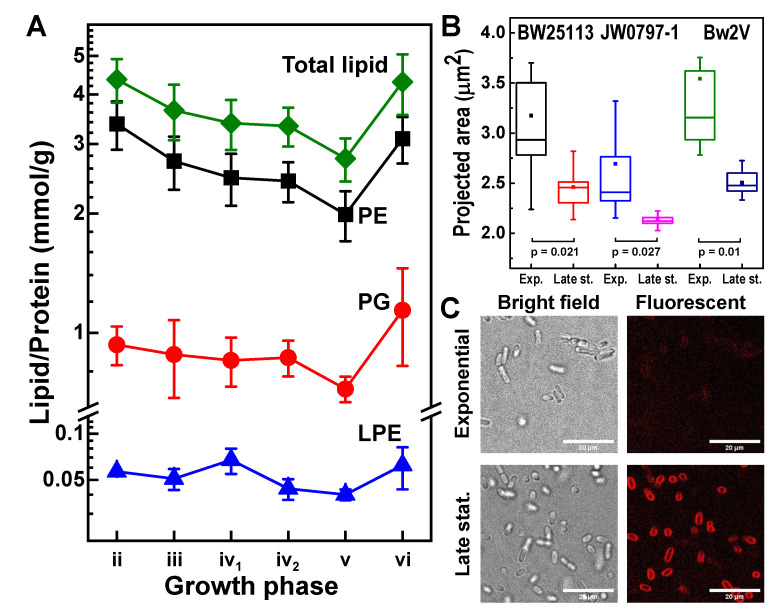
Lipid envelope and *E. coli* cell size dynamics. (**A**) The total lipid abundance based on the quantification of molecular species in BW25113 strain. (**B**) A larger projected cell areas in the exponential phase in comparison with the late stationary phase of growth in three strains of *E. coli*. Probability values are from a two-tailed two-sample equal variance *t*-test. (**C**) Micrographs of BW25113 strain in the exponential and the late stationary phases using bright field and fluorescent microscopies. The scale bar is 20 µm.

## Data Availability

The data presented in this study are available in Appendix A.

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
