# Peer review of "Transient Complexity of E. coli Lipidome Is Explained by Fatty Acyl Synthesis and Cyclopropanation"

_metabolites, 2022, doi:10.3390/metabo12090784_

Round 1

Reviewer 1 Report

In this manuscript, Nikolay V. Berezhnoy and colleagues provides a detailed quantitative lipidome analysis for the composition of three E. coli strains across culture growth phases, they analyzed the level of fatty acyl chains and a general reference for phospholipid composition changes in response to perturbations, e.g. antibiotics, that induced a nearly two-fold increase in saturated fatty acyl chains. The authors concluded that lipidome diversity is largely transient and the consequence of lipid synthesis and cyclopropanation. Overall, this manuscript is well written and qualified to be published in Metabolites, the authors, however, should consider the following specific comments to further strengthen the manuscript.

1\ In Figure 2A, it seems that the BM25113 data point (black) at 1h is missing? The author should explain the reason or add this data point.

2\ In Figure 3E, the authors should perform statistical calculation and calculate the relative p values between PE molecular species at different growth phases. And also other figures that involve comparison between different groups.

3\ In Figure 4E, based on the density of GAPDH bands in the WB results, the sample loading is significantly different, the author should repeat this WB experiments.

Reviewer 2 Report

The overall goal of this manuscript is to highlight changes in the lipidome composition by describing the dynamics of molecular species in E. coli strains throughout culture growth phases. The study provided a detailed quantitative lipidome composition of three E. coli strains across culture growth phases and at the level of fatty acyl chains and a general reference for phospholipid composition changes in response to perturbations.

·         The research is considered to be one of the first few studies done in the world that covers the quantitative lipidome dynamics during culture growth at molecular species-resolution that covers major glycerophospholipids classes of the E. coli lipidome.

·         The manuscript requires minor English language copyediting.

·         Please try to shorten the figure captions as much as possible. The captions as what they are, appeared to be overwhelming to read.

·         The first few paragraphs of the introduction are lengthy. I was only able to understand the main goal of the project until line 97, paragraph 6. I think emphasizing first on what is the main goal of the project would sustain capturing readers’ interest.

·         What is the overall importance and implication of studying the lipidome composition of E-coli strains in relation to humans? How will a study like this be beneficial to mankind? I think capturing the readers’ interest by positioning the writing from a general perspective and practical standpoint will go a long way.

·         Introduction, paragraph 7: Would it be “abundant” instead of “abundance” in the sentence as written as “Temporal resolution allowed us to identify abundance changes of individual molecular species,…”?

·         Under the introduction: I believe this sentence “The subsequent lack of composition at a level of molecular species, where FA chains are quantified, currently confounds the interpretation of exposure effects in response to drugs, organic solvents, or other kinds of lipidome stresses” should be further explicitly substantiated citing reliable and recent references.

·         Why is there a need to study the E. coli lipidome in a dynamic system during culture growth and as affected by environmental stresses?

·         How is this study different from other studies done in other cellular forms such as below?

o   Quantitative Analysis of the Cellular Lipidome of Saccharomyces Cerevisiae Using Liquid Chromatography Coupled with Tandem Mass Spectrometry - PubMed (nih.gov)

o   Absolute quantitative lipidomics reveals lipidome-wide alterations in aging brain | SpringerLink

o   Metabolites | Free Full-Text | Multi-Omic Analysis to Characterize Metabolic Adaptation of the E. coli Lipidome in Response to Environmental Stress (mdpi.com)

o   Monitoring Membrane Lipidome Turnover by Metabolic 15N Labeling and Shotgun Ultra-High-Resolution Orbitrap Fourier Transform Mass Spectrometry | Analytical Chemistry (acs.org)

·         How were the three strains of E. coli selected (i.e., basis of selection)?

·         What is the basis of selecting ampicillin and chloramphenicol?

·         Since the manuscript contains so many information, I think it is best that early on in the introduction, explicit information about what lipids will be quantified should be mentioned.

·         Would it be possible to provide a graphical abstract for this manuscript if it has not been submitted yet?

·         The manuscript is interesting. However, the paragraphs should have a coherence to be able to sustain the reader’s interest.

·         Was there a significantly different result on the amount of lipids of interest between Bw2V cultured in the presence of ampicillin and chloramphenicol?

Reviewer 3 Report

This article is reporting quantitative lipidome composition of three E.coli strains and concluded that lipidome diversity is largely transient as a consequence of lipid synthesis and cyclopropanation. Additionally, authors determined that lipid class compositions in the E.coli lipidome are consistent with specific ration of PE 69-77%, PG 21-29%, and LPE 1-2%. These numbers are with agreement of earlier published data. This will constitute the important goals and novelty of this paper.

            The following suggested changes and recommendations should be introduced before the publication of the manuscript.

1.     Page 3, line 106. The word “temporally” should be corrected to “provisionally” 

2.     Page 3, line 124. The sentence “ We expect this contribution” should be edited to the clearer format, as in the present wording is rather diluted. 

3.     Page 3, line 125.  The word “underpinning” should be replaced with “support”

4.     Page 6, line 257, 2.7. CFA synthase western blot assay, should have a literature reference. 

5.     Page 8, line 347. The title 3.3. Cyclopropanation dynamic of lipids“ should have literature reference [9]. 

The manuscript is of good quality and importance and is comprehensively written and edited in order to meet the standard for the review articles published in Metabolites. Thus, I certainly recommend it for publication after the correction of these suggested minor changes. 

Round 2

Reviewer 2 Report

All concerns have been addressed. This manuscript is now approved.